# The Open Cell Form of 3D-Printed Titanium Improves Osteconductive Properties and Adhesion Behavior of Dental Pulp Stem Cells

**DOI:** 10.3390/ma14185308

**Published:** 2021-09-15

**Authors:** Marialucia Gallorini, Susi Zara, Alessia Ricci, Francesco Guido Mangano, Amelia Cataldi, Carlo Mangano

**Affiliations:** 1Department of Pharmacy, University “G. d’Annunzio” Chieti-Pescara, 66100 Chieti, Italy; marialucia.gallorini@unich.it (M.G.); alessia.ricci@unich.it (A.R.); amelia.cataldi@unich.it (A.C.); 2Department of Pediatric, Preventive Dentistry and Orthodontics, Sechenov First State Medical University, 119991 Moscow, Russia; francescoguidomangano@gmail.com; 3Department of Dental Sciences, University San Raffaele, 20132 Milan, Italy; camangan@gmail.com

**Keywords:** titanium implants, biocompatibility, osteogenic differentiation, 3D printing, implant surface

## Abstract

Titanium specimens have been proven to be safe and effective biomaterials in terms of their osseo-integration. To improve the bioactivity and develop customized implants titanium, the surface can be modified with selective laser melting (SLM). Moreover, the design of macro-porous structures has become popular for reaching a durable bone fixation. 3D-printed titanium (Titanium A, B, and C), were cleaned using an organic acid treatment or with electrochemical polishing, and were characterized in terms of their surface morphology using scanning electron microscopy. Next, Dental Pulp Stem Cells (DPSCs) were cultured on titanium in order to analyze their biocompatibility, cell adhesion, and osteoconductive properties. All tested specimens were biocompatible, due to the time-dependent increase of DPSC proliferation paralleled by the decrease of LDH released. Furthermore, data highlighted that the open cell form with interconnected pores of titanium A, resembling the inner structure of the native bone, allows cells to better adhere inside the specimen, being proteins related to cell adherence highly expressed. Likewise, titanium A displays more suitable osteoconductive properties, being the profile of osteogenic markers improved compared to titanium B and C. The present work has demonstrated that the inner design and post-production treatments on titanium surfaces have a dynamic influence on DPSC behavior toward adhesion and osteogenic commitment.

## 1. Introduction

Dental caries and periodontitis affect millions of people worldwide and can have an irreversible impact on individuals causing severe tooth loss and diminished quality of life. Many options exist to replace missing teeth, however, the use of dental implants has become one of the most suitable strategies for replacing one or more missing teeth within recent decades. Contemporary dental implants made with titanium have been proven safe and effective in a large series of patients [1]. The surface properties of titanium and its alloys are commonly modified using different techniques, including additive manufacturing (AM), to improve the osseo-integration of dental implants and to develop customized implants with unrestricted geometries, controlled surface characteristics, and different porosity levels. Powder bed fusion (PBF) is the most popular metal AM method with bulk-supplied feedstock. In PBF, a high-energy electron beam (for electron beam PBF, known as PBF-EB), or a CO2/Nd:YAG/diode laser (for laser PBF, known as PBF-L), selectively fuse a bed of fine powder. PBF-L is widely referred to as selective laser melting (SLM), or selective laser sintering, depending on the degree of melting. Sintering (SLS) is used for nylons/polymers, whereas melting (SLM) for metal alloys, but in SLM, a complete melting process is performed instead of sintering [2,3]. Recently, macro-porous structures have become popular strategies for obtaining a durable bone fixation. Indeed, several studies have demonstrated that the implant surface topography, and an open interconnected porous structure with pores in the range of 200–400 µm, are required for bone regeneration and play a pivotal role in many peri-implant cellular and molecular mechanisms [4,5].

The osteoblastic differentiating potential of Dental Pulp Stem Cells (DPSCs) grown onto modified titanium surfaces has been widely demonstrated in several in vitro experiments [6,7,8]. As a matter of fact, the dental pulp contains progenitor stem cells with multi-lineage differentiation capabilities, which are able to generate an osteoblast-like phenotype when cultured in osteogenic-inducing media [9]. Therefore, DPSCs represent a safe, effective, and easily accessible cell source for regenerative bone purposes due to the demonstrated in vitro and in vivo expression of several typical early osteoblastic markers, such as alkaline phosphatase (ALP) and collagen type I, and late ones like osteopontin (OPN) and osteocalcin (OCN) when the cells are cultured on osteoconductive scaffolds and surfaces [10].

The conformation of titanium surfaces, as well as their roughness and their chemical properties, are also critical for cell adhesion and the presence of adhesion molecules can influence cell activities by interacting both with the implant surface and the extracellular matrix (ECM). It has been broadly demonstrated that the first biological reaction at the implant–tissue interface is the adsorption of body fluid proteins, including ECM components, onto the material surface after implantation [11]. For the cell–ECM interactions, the cell surface receptors are almost all integrins, and DPSCs are known to uniformly express the cell surface marker CD29 (Integrin β1) [12]. It has been recently reported that targeting integrins with agonist compounds on the surface of functionalized biomaterial scaffolds enhances the adhesion of human Mesenchymal Stem Cells (MSCs) by significantly driving changes in focal adhesion proteins distribution [13].

Interleukin-6 (IL-6) behaves as a pleiotropic cytokine in many cell processes, such as immune regulation, hematopoiesis, and in vivo tissue regeneration. In a study by Xie et al., the authors underlined that, under osteogenic induction, BM-MSCs are responsible for a continuous IL-6 secretion [14]. Our group has previously reported that a redox control is fundamental during the DPSC commitment to osteogenesis in the presence of biomaterials, and that this equilibrium is partly mediated by IL-6 secretion [15]. Furthermore, it has recently been reported that the production of IL-6 can be related to DPSC differentiation in the presence of materials designed for implantation. Moreover, pulp cells are capable of producing PGE2, which is mainly associated with angiogenesis and VEGF enhanced expression, representing key events during the regeneration process [16].

The present study aims at investigating the effect of suitable osteoconductive 3D-printed titanium specimens in terms of biocompatibility and of differentiation induction in in vitro cultured DPSCs.

## 2. Material and Methods

### 2.1. Synthesis of Titanium Specimens

The powder used for RenAM 500Q and MYSINT 100 printers was PowderRange Ti6-4 Grade 23 (Titanuim-6%, Aluminum-4%, Vanadium, Ti-6Al-4V) (produced by Carpenter Additive-Dennis Rd, Widnes WA8 0GU, UK). The powder was the highest purity version. PowderRange Ti6-4 Grade 23 is produced by plasma atomization, resulting in superior sphericity, low internal porosity, and low residual elements. The particle size is 10–20 µm.

Porous titanium specimens (type A-discs with 5 mm of diameter and 2 mm of depth, 200 specimens, Andrew Medical, Milan, Italy) were designed with an open cell form (interconnected pores) through SolidWorks^®^ 12.0 (SolidWorks Corp.Concord.MA, Waltham, MA, USA) and produced by a selective laser melting (SLM) machine (RenAM 500Q–Renishaw. The RenAM 500Q machine is equipped with four high power 500 W lasers, which are able to access the whole powder bed surface simultaneously. Its compact galvanometer assembly was designed and additively manufactured in-house, using aluminum for its high thermal conductivity, including conformal cooling fluid channels resulting in the excellent thermal stability of the optical system. The building parameters were a laser power of 200 W with a speed of 0.9 m/s, and a layer thickness of 15 µm. After that, the laser melted samples were vacuum heat-treated at 800 °C for 1 h. Type A underwent post-processing treatments by acid etching with organic acids. In order to remove the titanium spherical non-adherent particles from the surface, the samples underwent sonication for 5 min in deionized water at 25 °C, submerged in NaOH (20 g/L) and hydrogen peroxide (20 g/L) at 80 °C for 30 min, and then sonicated again for 5 min in deionized water. Following these processes, an organic acid treatment, in a 50% oxalic acid and 50% maleic acid mixture at 80 °C for 45 min, was applied to better clean the surface, followed by 5 min rinses with deionized water within a sonic bath, as previously reported [17].

Titanium specimens type B and C (discs with 5 mm of diameter and 2 mm of depth (200 specimens per type, BTK Biotec, Vicenza, Italy) were designed and produced without open cell form by SLM through a MYSINT 100-SISMA machine. The process parameters for the SLM method [3] were as follows: laser power, 50 W; scanning speed, 275 mm/s; and layer thickness, 20 µm. After the SLM process, the samples were heat-treated at 1200 °C for 1 h and allowed to cool at room temperature. As for the Ti specimens, type A, type B, and type C underwent post-processing manipulations. The type B samples were treated with organic acids, and also for the type A titanium discs. With regards to the type C specimens, they underwent electrochemical polishing, which is frequently used to reduce surface roughness by levelling its micro-peaks and valleys. The electrochemical polishing of the samples was performed using the Struers LectroPol-5 equipment. A solution containing 5% of perchloric acid (60%) in acetic acid was used as electrode, and a 0.5 cm^2^ mask for polishing of the top surface of the sample was selected. The samples were then polished at a room temperature of 25 °C for 5 min using a voltage of 55 V under a current of 0.3 A (Table 1) [18].

### 2.2. Cell Culture

Dental Pulp Stem Cells (DPSCs) were bought from Lonza (Lonza Group Ltd., Basel, Switzerland) and grown by using α-MEM medium (Sigma-Aldrich, St. Louis, MO, USA) added with 10% FBS, 1% penicillin/streptomycin (Euroclone S.p.A., Milan, Italy), at 37 °C and 5% CO_2_ up to passage 5.

DPSCs were seeded on each titanium disc and on round shaped Thermanox (Thermo Fisher Scientific™, Waltham, MA, USA) of the same density, based on the diameter of the latter, for 1, 3, 7, 14 and 28 days. In each well of a non-tissue culture treated 24 well plate (Falcon^®^, Corning Incorporated, New York, NY, USA) 1 mL of differentiation medium (DM) was added, or, rather, a complete α-MEM supplemented with osteogenic differentiating factors (dexamethasone10 nM, β‒glycerophosphate 5 mM and ascorbic acid phosphate 0.1 mM, all purchased from Sigma-Aldrich, St. Louis, MO, USA). At each established experimental time, supernatants were collected for further analyses.

### 2.3. Alamar Blue Cell Viability Assay

Viable DPSCs are able to reduce the alamarBlue reagent resazurin into its red product resurfin. At the set time point and for each titanium, the medium was replaced with a fresh one containing 10% of alamarBlue reagent (Thermo Scientific, Rockford, IL, USA), and afterwards incubated for 4 h at 37 °C. The absorbance was measured at 570 and 600 nm by means of a multiscan GO microplate spectrophotometer (Thermo Fisher Scientific). The value obtained without cells was established as negative control. The alamarBlue percentage of reduction was calculated according to the manufacturer’s instruction.

### 2.4. Lactate Dehydrogenase (LDH) Release Assay

For the purpose of quantifying the cytotoxic effect exerted on DPSCs by titanium discs, the CytoTox 96^®^ Non-Radioactive Assay (Promega Corporation, Fitchburg, WI) was performed. The assay quantitatively measures LDH, a stable cytosolic enzyme that is released upon cell lysis. Supernatants (50 µL) were pipetted in a 96 well plate with a flat bottom (Falcon^®^, Corning Incorporated, New York, NY, USA) and the volume was doubled adding the LDH reaction mixture. After 30 min of incubation at room temperature in the dark, 50 µL of stop solution were added and the optical density (O.D.) was measured at 490 and 690 nm and the obtained O.D. were normalized with the one related to alamarBlue.

### 2.5. Alizarin Red S (ARS) Staining

Calcium deposits, indicative of cell differentiation, can be detected with ARS. After 1, 3, 7, 14, and 28 days, each titanium disc was washed twice in PBS with Ca^2+^/Mg^2+^, and the monolayer of DPSCs adherent on the titanium was fixed in paraformaldehyde 4% for 15 min at room temperature, and then washed twice with distilled water. Then, 40 mm of Alizarin Red S solution (final concentration) (Sigma-Aldrich), was added to each titanium and probed at room temperature for 20 min on a shaker, followed by several rinses with distilled water in order to eliminate dye excess. For the spectrophotometric evaluation of calcium level deposits, they were made soluble by adding 800 µL/titanium of 10% acetic acid and then incubated for 30 min in agitation. Titanium discs were scraped and the sample containing calcium deposits was collected and vortexed in a tube. Hot mineral oil (Sigma-Aldrich) was added and the tube was kept on ice for 5 min and after that centrifuged at 20,000× *g* for 15 min. The supernatant was removed and 10% ammonium hydroxide was added, and the optical density of this solution was measured at 405 nm.

### 2.6. Alkaline Phosphatase (ALP) Activity

The alkaline phosphatase activity was evaluated by means of Alkaline Phosphate Assay Kit (Colorimetric) (Abcam, Cambridge, UK). This kit uses 50 µL of p-nitrophenyl phosphate (pNPP) 5 mM as a phosphatase substrate, which was added to 80 µL of each supernatant in duplicate and incubated for 1 h at room temperature in the dark. The reaction was stopped by adding 20 µL of stop solution in each well. The pNPP turned yellow when dephosphorylated by ALP, and the absorbance was read at 405 nm wavelength through a microplate reader (Multiskan GO, Thermo Scientific, Inc., Waltham, MA USA). The ALP activity (mU/mL/min) was calculated by following the manufacturer’s protocol.

### 2.7. ELISA Analysis of Collagen Type I and IL6

The amounts of collagen type I and IL-6 released in the supernatants for each time point were detected, respectively, using a Human Collagen Type 1 ELISA kit (Cosmo Bio Co., Ltd., Tokyo, Japan; cat. no. ACE-EC1-E105-EX) and IL-6 ELISA kit (Enzo Life Sciences, Farmingdale, NY, USA) following the manufacturer instructions. The absorbance was measured at 450 nm spectrophotometrically (Multiskan GO, Thermo Scientific, Waltham, MA, USA). The concentration of collagen type I (μg/mL) and IL-6 (pg/mL) was calculated using a standard curve generated with specific standards provided by the manufacturers and normalized with the alamarBlue values.

### 2.8. Hydrogen Peroxide Release

The concentration of hydrogen peroxide was estimated through the Hydrogen Peroxide colorimetric detection kit (Enzo Life Sciences, Inc., Farmingdale, NY, USA; cat. no. ADI-907-015) in the same cell supernatants previously described. Then, 50 µL of each sample were pipetted in a well of a Half-Area Microtiter Plate in duplicate, and 100 µL/well of color reagent was added and wells were further incubated for 30 min at room temperature. The optical density (OD) was measured at 550 nm using a spectrophotometer (Multiskan GO; Thermo Fisher Scientific, Inc., Waltham, MA USA). The results were calculated by subtracting the average blank O.D. from the average O.D. measured for each sample, and then by interpolating the obtained values with a standard curve, following the manufacturer’s specifications.

### 2.9. Cell Lysis and Protein Extraction

After 1, 14, and 28 days of culture, DPSCs were trypsinizated and scraped from the titanium specimens. Next, samples were collected in complete α-MEM by centrifugation and pellets were then rinsed with cold PBS. Lysis buffer (50 µL) with a protein inhibitors cocktail (PBS, 1% IgePal CA-630, 0.5% sodium deoxycholate, 0.1% SDS, 10 mg/mL PMSF, 1 mg/mL aprotinin, 100 mM sodium orthovanadate and 50 µg/mL leupeptin), was added to the pellets, and samples were then maintained on ice for 30 min, re-suspended, and kept on ice for additional 30 min. Cell lysates were centrifuged for 15 min at 20,000× *g* and supernatants with proteins were collected. The protein concentration was measured through a bicinchoninic acid assay (QuantiPro™ BCA Assay kit for 0.5–30 µg/mL protein, Sigma-Aldrich, Milan, Italy) following the manufacturer’s indications.

### 2.10. Immunoblotting

Cell lysates (10 µg/sample) underwent electrophoresis on a 4–20% SDS-PAGE Gel (ExpressPlus™ 10 × 8, GenScript Biotech Corporation, Nanjing, China) and then were transferred to nitrocellulose membranes. The membranes underwent saturation in 5% of non-fat milk or 5% of BSA, 10 mmol/L Tris pH 7.5, 100 mM NaCl, 0.1% Tween 20 and incubated overnight at 4 °C under light agitation in the presence of mouse monoclonal anti-β-actin (Sigma-Aldrich, St. Louis, MO) (antibody dilution 1:10,000), mouse monoclonal anti-Integrin β1 (antibody dilution 1:200), rabbit polyclonal anti-VEGF (antibody dilution 1:200) (bought from Santa Cruz Biotechnology, Santa Cruz, CA, USA), rabbit anti-p44/42 MAPK, anti-phosho-p44/42 MAPK (Erk 1/2 and p-Erk 1/2) monoclonal antibodies (primary antibodies dilution 1:1000) (all purchased from Cell Signaling Technology, Danvers, MA, USA), and mouse monoclonal anti-FAK (primary antibody dilution 1:500) (purchased from ThermoFisher Scientific, Waltham, MA, USA). Subsequently, the membranes were probed in the presence of specific IgG horseradish peroxidase (HRP)-conjugated secondary antibodies. Immuno-reactive bands were revealed using the ECL detection system (LiteAblot Extend Chemiluminescent Substrate, EuroClone S.p.a., Milan, Italy) and underwent densitometry. Densitometric values, expressed as Integrated Optical Intensity (I.O.I.), were estimated in the CHEMIDOC XRS system using the QuantiOne 1-D analysis software (BIORAD, Richmond, CA, USA). The values obtained were normalized based on densitometric values of internal β-actin.

### 2.11. Scanning Electron Microscopy (SEM) Analysis

SEM analyses were then performed to evaluate the relationship between DPSCs and the titanium disc surfaces. Cells were fixed on the titanium with 1.25% glutaraldehyde in 0.1 M cacodylate buffer at pH 7.2 for 30 min. Subsequently, they were dehydrated with alcohol series and dried with hexamethyldisilazane. Images were obtained at 15 kV with a Phenom XL SEM microscope (Thermo Fisher Scientific, Eindhoven, Netherlands).

### 2.12. Statistics

The statistical analysis was established by the analysis of variance (one-way ANOVA), followed by Tukey’s post-hoc test, both performed with the Prism 5.0 software (GraphPad, San Diego, CA, USA). The results were represented as means ± SD. Values of *p* < 0.05 were considered statistically significant.

## 3. Results

DPSCs were cultured in a differentiation medium on three titanium surfaces, named titanium A, B, and C, up to 28 days. Titanium specimens underwent physical modifications in order to improve their biocompatibility and performance.

### 3.1. SEM Morphological Analyses

SEM analysis was carried out on titanium surfaces A, B and C before cell seeding and then after 1, 7, 14, and 28 days of culture to evaluate DPSC morphology and adhesion. Titanium laser melted sample (Titanium A) shows the porous structure, typical of the open cell form, with interconnected pores. Moreover, as the treatment with organic acids remove all non-melted particles, the higher magnification of titanium A image discloses some particles fused with surface laser. The melted sample B (titanium B) reveals the characteristic aspect of homogeneous concavities, due to the treatment with organic acids, whereas Titanium C represents a laser melted material which underwent electropolished surface treatment, thus showing a smoother surface (Figure 1).

SEM analysis performed after cell seeding evidenced that DPSCs are able to adhere and to proliferate on all the tested titanium surfaces in a time dependent manner; DPSCs extend their cytoplasmic extensions creating a thick network all over the available surface (Figure 2). In particular, on titanium disc A, DPSCs adhere on the surface and are also able to infiltrate within the pores; from day 1 to day 28, cells acquire their definitive shape becoming more and more flat, spreading on all the titanium surface and thus creating a cell monolayer. Moreover, on titanium A, after a long period of culture (day 28), it is possible to identify the deposition of inorganic matrix granules. On the B and C titanium surfaces, a similar situation can be identified even if, being that these samples lacking of large pores, cells are all kept on the disc surface (Figure 2).

### 3.2. Viability and Cytotoxicity Evaluation

To evaluate DPSC proliferation, the alamarBlue assay was performed at all the established experimental times. For all the experimental conditions, the metabolic activity increases starting from 1 up to 14 days, while after 28 days a slight reduction is registered (Figure 3A).

After 1 day of culture, the proliferation rate appeared to have increased when DPSCs were cultured on coverslips (control) with respect to titanium A, B and C, and an augmented cell proliferation on titanium B compared to titanium A was also detectable. After 7, 14, and 21 days of culture, the same time-dependent trend was maintained, and the cell proliferation rate recorded for titanium C appears to have statistically increased with respect to titanium A. After 28 days of culture, the proliferation level registered for titanium B and C, and for the control sample, are statistically augmented compared to the one detected for titanium A.

Next, cytotoxicity occurrence was evaluated by measuring the LDH released within the culture medium (Figure 3B).

After both 1 and 3 days of culture, a statistically significant peak in LDH release was detected for the control sample with respect to all the three tested titanium surfaces, and also for the titanium B compared to the titanium A. Starting from 7 days of culture, the recorded differences between samples appeared less evident; in particular, a significant reduction in the LDH released by cells on titanium A, with respect to the other tested surfaces, can be measured after 7 days of culture, whereas no differences were evidenced after 14 days. Finally, LDH released from DPSCs grown onto titanium B and C for 28 days raised dramatically when compared to the amount released in the presence of titanium A.

### 3.3. Adhesion Parameters Analysis

The DPSC adhesion capability was investigated by considering the protein expression of FAK and β1-integrin along with Collagen type I secretion within the culture medium corroborated by SEM morphological observations (Figure 4).

FAK and β1-integrin protein expression was measured through western blot analysis performed after 1, 14, and 28 days of culture (Figure 4A). After 1 day, β1-integrin appears to have expressed only in cells cultured on titanium B, while after 14 days of culture protein levels were significantly higher in the presence of titanium A compared to B, and in titanium B compared to C. After 28 days of culture, differences in the β1-integrin expression-levels cannot be evidenced. On the other hand, FAK protein expression was prominently reduced in the presence of titanium B compared to titanium specimens A and C after 1 and 14 days of culture, whereas a statistically significant increase in protein expression can be measured for titanium B and C respect to titanium A after 28 days of culture (Figure 4A).

The secretion of type I Collagen was evaluated using the ELISA assay. Notably, a consistent peak of secretion was recorded for titanium A after 3 days with respect to all the other tested conditions. After 7 days of culture, Collagen I secretion raises up significantly when cells were cultured on titanium specimens A and C, with respect to titanium B and the control. A similar trend can be observed after 14 and 28 days of culture, being that the collagen type I secretion recorded on titanium A was higher than the one measured for all the other tested samples (Figure 4B). The morphological analysis performed using SEM confirmed the previous result showing a better organized network of collagen fibers on titanium A, compared to what can be observed for titanium B and C after 1 day of culture (Figure 4C).

### 3.4. Extracellular Matrix Deposition Measurement

In order to evaluate the extracellular mineralized matrix deposition, Alizarin Red Staining was performed. At early experimental times (days 1 and 3), matrix deposition appeared to be significantly enhanced in the control samples (performed on polystyrene coverslips) compared to all the other tested conditions, and in the presence of titanium B with respect to titanium A and C. After 7 days of culture, the maximum level of deposed matrix was recorded for titanium A and compared to titanium B and C. Additionally, a substantial increase in the matrix deposition was also recordable for titanium B with respect to titanium C. Likewise, after 14 days of culture, cells cultured on titanium A were able to deposit a larger amount of mineralized extracellular matrix compared to cells grown on titanium B and C. To conclude, after 28 days a higher matrix deposition was detectable for titanium A with respect to B and C, even if there was no statistical significance. Nevertheless, a statistically significant decrease of matrix deposition for titanium B and C can be recorded in comparison to the control (Figure 5A).

### 3.5. Interplay between Osteoblastic Differentiation and Oxidative Stress

The DPSC differentiation process towards the osteoblastic phenotype was checked by measuring the ALP activity. In general, ALP activity decreases time-dependently and in all the experimental conditions. Indeed, the highest activity of the ALP enzyme can be detected after 1 day for all the experimental samples. Cells onto titanium A and C exhibit the highest levels of ALP activity over the experimental times, even though these differences are not statistically significant (Figure 5B).

With the aim of evaluating the DPSC redox state, the release of intracellular H_2_O_2,_ along with the IL-6 secretion levels, were estimated. At the earliest experimental times (days 1 and 3), a peak in H_2_O_2_ release can be highlighted in DPSCs cultured on titanium A compared to the ones grown onto titanium B and the control sample (day 1), and compared to all the experimental conditions after 3 days of culture. After 7 days, the highest level of H_2_O_2_ release is detected for cells cultured on titanium C respect to all the experimental conditions. After 28 days, a notable increase in H_2_O_2_ release is evident in the control with respect to all the titanium discs (Figure 6A).

The expression levels of activated Erk (p-Erk) and VEGF proteins were evaluated using a western blot analysis after 1, 14, and 28 days. After 1 day of culture, p-Erk expression was detectable only for titanium B, while after 14 and 28 days activated Erk raises statistically in the presence of titanium A with respect to titanium B and C (Figure 6B).

To conclude, IL-6 secretion within the culture medium and VEGF expression levels were measured. IL-6 secretion levels appeared to be significantly augmented in DPSCs cultured on coverslips (control) and on titanium B respect to titanium A after 3 days of culture whereas, after 7 and 14 days, a statistically significant peak in IL-6 secretion can be evidenced on titanium A compared to titanium B (7 days) and to the control (14 days). After 28 days of culture, a statistically significant peak of IL-6 secreted from DPSCs on titanium C is registered in comparison to all other experimental points (Figure 7A).

Moreover, cells onto titanium B show a IL-6 secretion rate significantly increased compared to the ones on titanium A. VEGF expression is not recorded after 1 day of culture for all the experimental conditions, while it appears to be statistically reduced in DPSCs cultured on titanium C with respect to the ones grown onto A and B. Moreover, the levels of VEGF were reduced in the presence of titanium A compared to titanium B after 14 days of culture. Contrariwise, a significant increase in VEGF protein level was recorded in cells grown on titanium C compared to titanium A and B after 28 days (Figure 7B).

## 4. Discussion

Implant morphology represents a strategic element in bone-implant interaction and can promote and accelerate the process of osseo-integration. In order to ameliorate dental implant stability, several surface modifications have been studied to adapt titanium dental implants properties. Morphological or chemical treatments, such as surface roughening, can be responsible for modifications in the chemistry of the dental implant surface [19]. In particular, surface topography and roughness can be easily manipulated by resorting to post-production surface treatments, and can play a pivotal role in the biological response influencing cell adhesion, adsorption, and differentiation.

In general, the mechanical and biological performance of the porous metal structure is governed by the combined effects of characteristic features of the porosity, such as pore shape, size, distribution, and their interconnectivity.

Selective laser melting (SLM) technology, in which the small metal powder particles are melted and fused in layers by a laser, is one of the metal additive manufacturing techniques which can directly produce implants following the complex 3D models design using CAD/CAM computer methods This technology enables one to simultaneously fabricate the porous structure and the main body of the implant, as well as personalized implants with complex structures fitted to the specific defect of the patient [4,20]. As the materials are produced from metal powders, SLM materials characteristically have very rough surfaces.

In our experimental model, we used porous SLM titanium (type A discs) designed with an open cell form (interconnected pores) (Appendix A), and Dense SLM titanium (types B and C discs), designed without an open cell form and produced to obtain a full density. Laser melted samples underwent different surface post-processing polishing manipulations: type B samples was treated with organic acids, whereas type C specimens underwent electrochemical polishing, which is frequently used to reduce the surface roughness by levelling micro-peaks and valleys.

Thus, the above-mentioned titanium were here biologically characterized aiming at evaluating whether the different surface treatments applied could better and faster promote DPSCs differentiation towards the osteoblastic phenotype.

DPSCs were chosen because they represent a valuable cell source for hard tissues regeneration [21] and for their appreciable differentiation ability [22].

The osteogenic commitment of MSCs is characterized by a high proliferation rate and a change in cell energy metabolism. It has been reported that many types of stem cells rely on glycolysis for energy when undifferentiated, and then later activate the heavily mitochondrial process of oxidative phosphorylation (OxPhos) during differentiation leading to a highest metabolic state [23]. This is in line with our experimental data related to the alamarBlue assay up to 21 days, where DPSC proliferation is clearly increased. However, after 28 days, the proliferation of differentiated cells slightly decreases. As broadly known, osteoblasts and osteoblast-like cells are mature and highly differentiated post-mitotic cells whose function is to produce the mineralized matrix, and are thus characterized by a low proliferation rate and an augmented protein synthesis. This trend has been already reported by our group relating to established cell/biomaterial experimental models in the presence of DPSCs [16]. The biocompatibility analysis clearly shows a completely different cytotoxic effect of the three surfaces evidencing for titanium A, such as a very low profile of toxicity, which appears to be maintained all the experiment long (28 days). This data appears even more appreciable when noticing that the extremely low release of LDH is also recorded at the very early stages of culture period when, generally, DPSCs are used to release large amounts of LDH even in presence of the most promising biomaterials [7]. In this context, SEM analysis underlines deep differences between titanium surfaces and, as a consequence, a different DPSC adhesion and spreading. Titanium A is designed in order to accurately mimic the native bone tissue structure, thus, huge pores are detectable on the surface A, whereas they are completely missing on titanium B and C. At the early stages of DPSC culture, it is possible to notice that the cells are able to infiltrate and to adhere within the pores of surface A, filling them and thus forming a continuous cell layer only after at least 14 days of culture. Conversely, having titanium B and C non-porous surfaces, all DPSCs adhere early (7 days) on the top of the discs, thus creating a cell layer throughout the surface. These data are perfectly in alignment with the alamarBlue viability results, in which the reduced cell viability rate recorded on titanium A, with respect to B and C, can be attributable to the fact that lots of DPSCs grow up and proliferate within the pores of the disc, thus not contributing to the alamarBlue-detected signal.

The DPSC varied modulation of proteins related to cell attachment and proliferation reflects the observed dissimilar behaviour of adhesion and growth ascribable to the different titanium surfaces. The collagen-binding integrin β1 is broadly known as a regulator of osteoblast differentiation and proliferation on titanium substrates with different roughness characteristics [24], and it has been found to be time-dependently increased during the osteogenic commitment of DPSCs [25]. On the other hand, focal adhesions (FAs) act as a bridge between integrin-ECM connection and the cytoskeleton, and are required for morphogenesis and migration [26]. It has been recently reported that a potential role for the Integrin α₅β₁/Focal Adhesion Kinase (FAK) is signaling mechanotrasduction and collagen synthesis in the presence of scaffolds used in orthodontic treatment [27]. In our experimental model, Integrin β1 and FAK expression appears inversely modulated. Once activated, FAK binds to cell membrane integrins, resulting in focal contacts and cell adhesion. The highest amount of this protein in the presence of titanium A, compared to the other specimens at the earliest experimental time (1 day), could be a signal of the strong cell adhesion inside the interconnected porous structure. Moreover, DPSCs grown onto/into specimen A show a lower amount of FAK at day 28 with respect to titanium B and C. Data reported in literature demonstrated that FAK drives cell adhesion and survival, and cytoskeleton formation in the dental pulp, but an excessive and prolonged FAK expression has no positive effects on these aspects [28]. The reduced expression of FAK can be in accordance with cytotoxic cells response to titanium A, which is shown to be highly biocompatible due to the small amount of LDH released over all the experimental exposures. Additionally, a peak in collagen type 1 production is registered after 3 days of culture in the presence of titanium A, a signal widely recognized to be crucial in pre-osteoblast adhesion on three dimensional biomaterials and surfaces which involve the binding to β1 integrins [29]. A tightly physiological upregulation of ROS is required for MSC differentiation, and it has been reported that WNT/β-catenin activation increases ROS during osteogenesis [30]. However, elevated oxidative stress due to the accumulation of hydrogen peroxide (H_2_O_2_) lead to MSC cell cycle arrest and apoptosis [31]. It has been reported that ROS mediate the oxidation of Cys124 in extracellular signal-regulated kinase (ERK), an action that results in its activation by phosphorylation, as a signal of cell survival [32]. It is plausible to assume that the extremely high amount of H_2_O_2_ produced in the presence of titanium A at day 1, and persistent at day 7, is then counteracted by the activation of Erk after 14 days which is maintained also at day 28. Additionally, intracellular pathways, where ROS and protein related to osteogenesis converge, have been described, and the activation of the Erk signaling has been reported among them [30]. It is not therefore surprising that the early H_2_O_2_ burst in the presence of titanium A corresponds to a better osteogenic marker profile with respect to specimens B and C, in terms of persistent ALP activity up to day 28 and biomineralized matrix deposition, the latter also confirmed by SEM analysis, in which crystals of hydroxyapatite are clearly detectable only on titanium A surface. Moreover, the alizarin red-related signal after 7 days with Titanium A is perfectly in alignment with the dramatic peak recorded for Collagen I release, indicating the subsequent mineralization of the organic matrix.

Being the PGE2-mediated synthesis of IL-6 and VEGF in osteoblasts highly involved in the bone metabolism, as previously demonstrated [33], the analysis of these two factors trend shed light on the fact that at later stages the surface of Titanium C triggers in DPSCs typical bone remodeling pathways, but these molecular events are not supported neither by a complete osteoblastic differentiation, nor by inorganic matrix deposition.

## 5. Conclusions

In the present paper, laser melted titanium specimens were produced with CAD drawn open cell form or without (dense form) undergoing different post-production treatments to remove particles powder adhered to the surface (i.e., with organic acids or electrochemical polishing). All specimens were characterized in terms of their surface morphology. Furthermore, their biocompatibility and osteoconductive features were investigated in vitro in the presence of mesenchymal cells from dental pulp (DPSCs). Considering the viability and cytotoxicity data obtained, all the three specimens are biocompatible and are definitely suitable for cell adhesion and proliferation processes. A better profile in terms of markers related to an early triggered cell osteogenic commitment is found mainly in titanium A, along with an ameliorated cell behaviour, plausibly ascribable to its open cell inner structure resembling the native bone trabecular structure. Likewise, proteins involved in cell adhesion and survival, as well as the cytokines released, are differentially modulated among the three different specimens, confirming the dynamic influence of the inner design and of post-production treatments on DPSC cell response and interaction toward titanium surfaces. The porous structure of titanium drawn in CAD and produced by laser melting (SLM) demonstrates greater bioactivity than dense surfaces.

## Figures and Tables

**Figure 1 materials-14-05308-f001:**
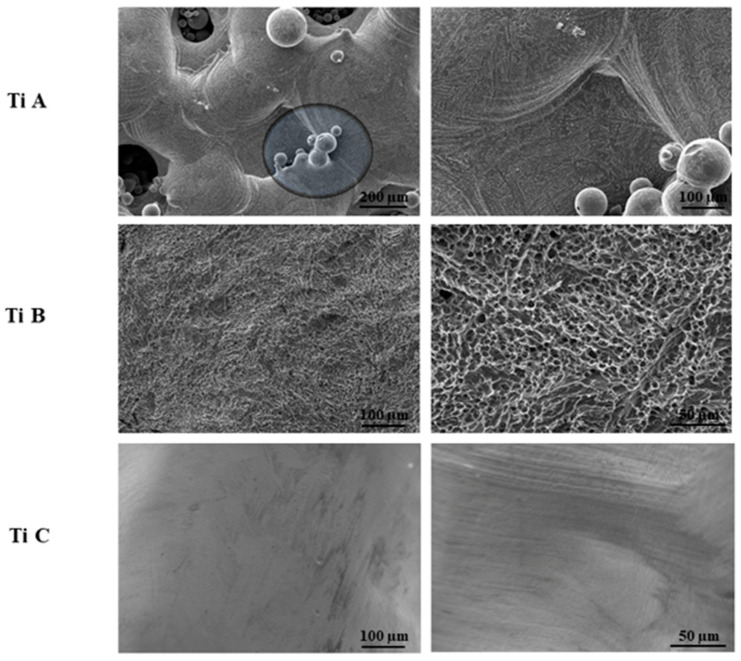
SEM images of Titanium A, Titanium B and Titanium C surfaces. Magnification: Titanium A 200× left image, 500× right image, Titanium B and C 500× left image, 2000× right image.

**Figure 2 materials-14-05308-f002:**
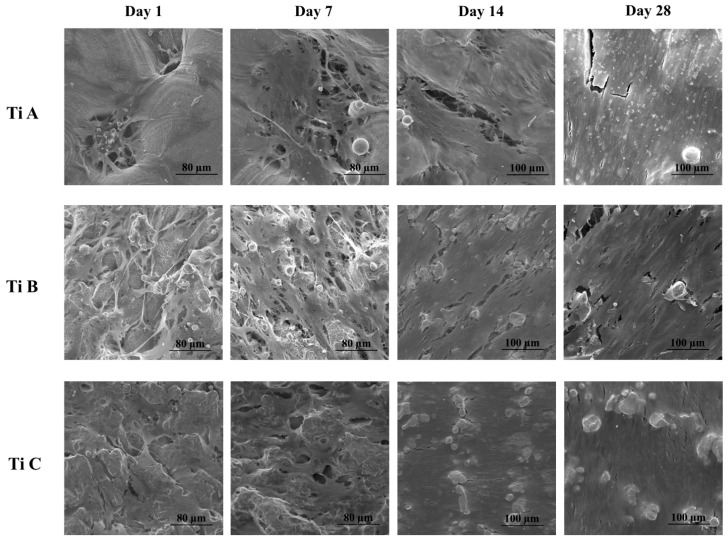
SEM images of DPSCs cultured on Titanium A, Titanium B and Titanium C for 1, 7, 14 and 28 days. Magnification: Day 1 and day 7 900×; day 14 and day 28 600×.

**Figure 3 materials-14-05308-f003:**
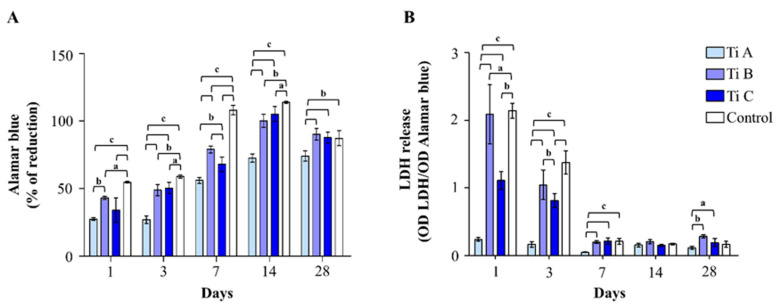
(**A**) The alamarBlue assay in DPSCs cultured on Titanium A, B, C and coverslip scaffold (control) for 1, 3, 7, 14, and 28 days. The histogram represents the % of alamarBlue reduction values. Data are the mean (±SD) of three different experiments. (**B**) LDH assay of DPSCs cultured on Titanium A, B, C and coverslip scaffold (control) for 1, 3, 7, 14, and 28 days. LDH released is reported as OD LDH/OD alamarBlue ratio. Data shown are the mean (±SD) of three separate experiments. a = *p* < 0.01; b = *p* < 0.001; c = *p* < 0.0001.

**Figure 4 materials-14-05308-f004:**
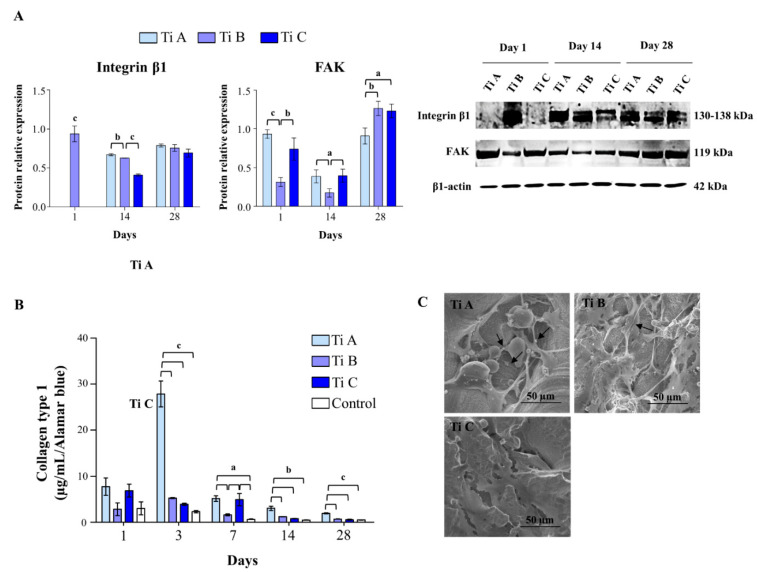
(**A**) Western blotting analysis of Integrin β1 and FAK expression in DPSCs grown on Titanium A, B and C for 1, 14 and 28 days. Each membrane has been incubated with β1–actin antibody to verify loading consistency. Western blot is the most representative of three different experiments. Histograms represent densitometric measurements of proteins bands expressed as integrated optical intensity (IOI) mean of three different experiments. The error bars show standard deviation (±SD). (**B**) ELISA assay for Collagen type I secretion of DPSCs cultured on Titanium A, B, C and coverslip scaffold (control) for 1, 3, 7, 14, and 28 days. Secretion levels are reported as µg per milliliter per alamarBlue values. The results are the mean ± SD of three samples from three different experiments. (**C**) SEM morphological analysis performed on DPSCs cultured on Titanium A, B and C after 1 day., black arrows indicate collagen fibers; magnification 1450× (Ti A), 1150× (Ti B) and 1550× (Ti C). a = *p* < 0.01; *b* = *p* < 0.001; c = *p* < 0.0001.

**Figure 5 materials-14-05308-f005:**
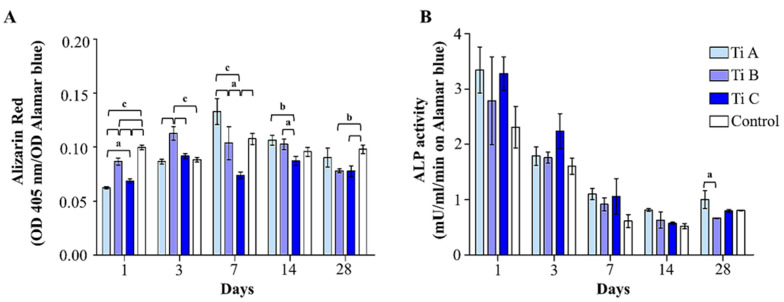
(**A**) Alizarin red staining representing OD values of solubilized orange-red stained calcium deposits/OD alamarBlue ratio. Data shown are the mean (±SD) of three separate experiments. (**B**) Quantitative evaluation of the ALP activity in DPSCs grown onto Titanium A, B, C and coverslip scaffold (control) for 1, 3, 7, 14, and 28 days. Values represent the means ± SD. Bar graph showing the enzymatic activity of ALP (U/mL) normalized on alamarBlue values. a = *p* < 0.01; b = *p* < 0.001; c = *p* < 0.0001.

**Figure 6 materials-14-05308-f006:**
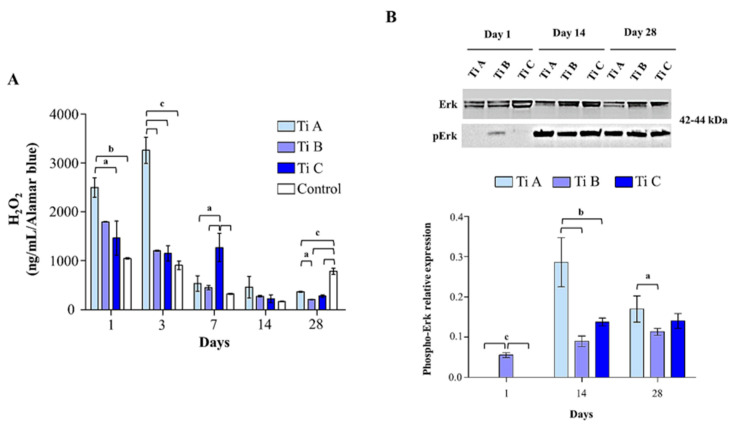
(**A**) Quantitative evaluation of the H2O2 released in DPSCs grown onto Titanium A, B, C and coverslip scaffold (control) for 1, 3, 7, 14, and 28 days. Values represent the means ± SD of three separate experiments. The bar graph shows the H2O2 release (ng/mL) in cell supernatants normalized on alamarBlue values. (**B**) Western blotting analysis of activated Erk in DPSCs cultured on Titanium A, B and C for 1, 14 and 28 days. Each membrane has been incubated with β1–actin antibody in order to verify loading consistency. Western blot is the most representative of three different experiments. Histograms show densitometric measurements of proteins bands expressed as integrated optical intensity (IOI) mean of three different experiments. The error bars show standard deviation (±SD). a = *p* < 0.01; b = *p* < 0.001; c = *p* < 0.0001.

**Figure 7 materials-14-05308-f007:**
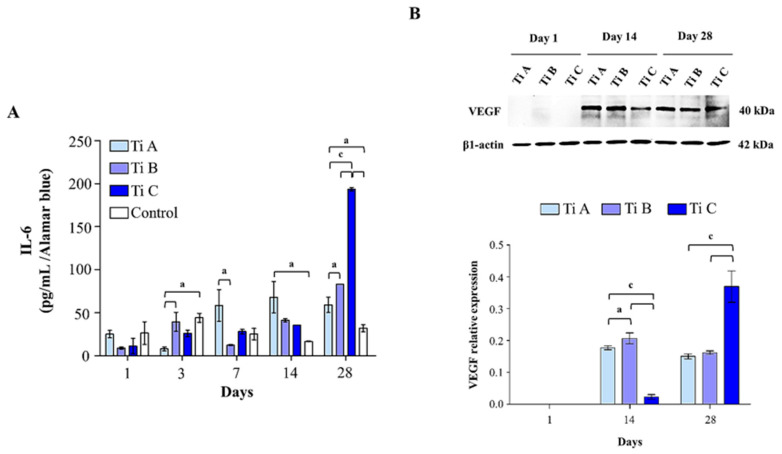
(**A**) ELISA assay for IL-6 secretion of DPSCs cultured on Titanium A, B, C and coverslip scaffold (control) for 1, 3, 7, 14, and 28 days. Secretion levels are reported as µg per milliliter per alamarBlue values. The results are the mean ± SD of three samples from three different experiments. (**B)** VEGF expression in DPSCs cultured on Titanium A, B and C for 1, 14 and 28 days. Each membrane has been probed with β1–actin antibody to verify loading consistency. Western blot is the most representative of three different experiments. Histograms represent densitometric measurements of proteins bands expressed as integrated optical intensity (IOI) mean of three separate experiments. The error bars show standard deviation (±SD). a = *p* < 0.01; b = *p* < 0.001; c = *p* < 0.0001.

**Table 1 materials-14-05308-t001:** Samples parameters and applied post-processing procedures.

Header	Material	Size/Shape	Synthesis	Heat Treatment	Post-Processing
Type A	Ti6Al4V	Discs 5 mm diameter and 2 mm depth with open cell form	RenAM 500Q (Renishaw) Laser power 200 W Speed 0.9 m/s Layer thickness 15 µm	Vacuum heat treatment at 800 °C for 1 h	Sonication for 5 min in deionized water at 25 °C, submerged in NaOH (20 g/L) and hydrogen peroxide (20 g/L) at 80 °C for 30 min, and then sonicated again for 5 min in deionized water. Organic acid mixture (50% maleic/oxalic) at 80 °C for 45 min.
Type B	Ti6Al4V	Discs 5 mm diameter and 2 mm depth without open cell form	Mysint 100 (Sisma) Laser power 50 W Speed 275 mm/s Layer thickness 20 µm	Vacuum heat treatment at 1200 °C for 1 h	Sonication for 5 min in deionized water at 25 °C, submerged in NaOH (20 g/L) and hydrogen peroxide (20 g/L) at 80 °C for 30 min, and then sonicated again for 5 min in deionized water. Organic acid mixture (50% maleic/oxalic) at 80 °C for 45 min.
Type C	Ti6Al4V	Discs 5 mm diameter and 2 mm depth without open cell form	Mysint 100 (Sisma) Laser power 50 W Speed 275 mm/s Layer thickness 20 µm	Vacuum heat treatment at 1200 °C for 1 h	A solution containing 5% of perchloric acid (60%) in acetic acid was used as electrode, and a 0.5 cm^2^ mask for polishing of the top surface of the sample was selected. The samples were then polished at room temperature of 25 °C for 5 min using a voltage of 55 V under a current of 0.3 A

## Data Availability

Not applicable.

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
