# Peer review of "The Open Cell Form of 3D-Printed Titanium Improves Osteconductive Properties and Adhesion Behavior of Dental Pulp Stem Cells"

_materials, 2021, doi:10.3390/ma14185308_

Round 1

Reviewer 1 Report

The work "The open cell form of 3D-printed titanium improves osteconductive properties and adhesion behavior of Dental Pulp Stem Cells " is devoted to using of titanium additive manufacturing of biocompatible dental implants.

  1. Abstract:
    line 15 "improve" what do you mean here by improving? More biocompatible, more smooth, rougher? 
    line 16 - please remove "and sintering (SLS)"
    lines 15-16 and lines 18-19 - please decide how you have modified the surface by means of printing or by means of treatment.
  2. In Fig.1 the scale  bar is missing.
  3. In Fig.2  and Fig.4c the scale bars are not readable, please fix that.
  4. lines 43-44 "Among the methods of AM, selective laser melting (SLM) and sintering (SLS) have been 43 the predominant methods for the AM of titanium alloys." - please provide a reference for this statement. 
    First, what about Electron Beam melting?
    Second, SLS and SLM are the same technology. The main difference between them is the fusing process - or sintering or melting. Sintering (SLS) is used for nylons/polymers; melting (SLM) - for metal alloys. The term "SLS" is a historical name of the same powder bed technology. Now, for titanium alloys, only SLM is used, not SLS.
  5. Please, fix this part in Introduction (lines 43-48). I also recommend to observe some relevant papers on this topic, e.g.:
    https://doi.org/10.1007/s13534-018-0080-5 https://doi.org/10.1016/j.matdes.2017.07.059
    https://doi.org/10.1007/s13534-018-00092-7 etc.
  6. line 71 "MSCs" - What does it mean? Please give an explanation of all abbreviations.
  7. lines 78-79 "Furthermore, we have recently reported..." - you can't write like that since there were other authors in [16].
  8. lines 91-93 "The additive manufacturing (AM) SLM technology is based on the process called metal powder bed fusion in which material powders are fused together [17]." - IT is not relevant for Methods and Materials Section. Please, or remove, or put that in Introduction.
  9. line 99 "After that, the laser melted samples where heat-treated at 800°C for 1 hour" - in what environment? Vacuum/Argon?
  10. Section 2.1. synthesis of the samples types A, B, and C. I recommend transforming this section into a representative table indicating process parameters, and applied post-processing procedures.
    Type A - Material - Size/shape - sinthesis - heat treatment - post-processing
    Type B ....
    Type C ...
    Just a short explanation is required why you have used different printing machines for different types of samples?  And why do you show different process parameters e.g. layer thickness is shown only for the Sisma machine?
  11. Please, specify, the powder data - supplier, morphology, particle size dispersion, production technique, chemical composition, etc.
  12. line 412 - just "melted", no need to say each time "melted and fused".
  13. lines 411-418, please try to avoid such characteristics as "small powders" or "very rough surface". These characteristics are measurable. The standard size range for SLM powders is 25-50 microns. The standard roughness is related to the powder you use, so for SLM it also will be of the order of 50 microns. Anyway, this paragraph is more suitable for Introduction not for a discussion.
  14. line 116 and 424 "surface roughness" for which purpose you put a link on Wikipedia here???
  15. lines 419-425 - it is repeated information, please remove.
  16. line 426 "was" instead of "were"
  17. Conclusion: "On the other hand, pulp cells grown on type A titanium show a better profile in terms of markers related to osteogenesis, a cell behaviour plausibly ascribable to its open cell inner structure, which resembles the native bone trabecular structure." Please, explain why it is preferable from your point of view to treat the surface of titanium implants, instead of just printing from the beginning the trabecular surface [https://doi.org/10.1007/s13534-018-00092-7]?
  18. I recommend to strengthen the Conclusion and maybe present the main findings in the form of a bullet list.
  19. The paper has serious gaps, and I recommend to work on Abstract, Introduction summarizing part, and on Conclusion, to formulate the novelty and value of the presented research.
  20. I also don't understand the classification of the investigated samples: A - titanium with porous structure; B and C without. Moreover, you apply different treatments to A and to B/C. That means you simultaneously compare the effect of porosity and treatment, and it is not good. More reasonable to compare samples type A as-built, heat-treated, and post-processed with different cell structures. Thus, you can compare the initial biocompatibility of the additively manufactured porous titanium and the biocompatibility of the samples with a treated surface. Otherwise, your experiment design is not clear.

Author Response

Abstract:

  • line 15 "improve" what do you mean here by improving? More biocompatible, more smooth, rougher?

Answer: we mean to improve the bioactivity of the surface. We added the explanation within line 15.

  • line 16 - please remove "and sintering (SLS)"

Answer: corrected

  • lines 15-16 and lines 18-19 - please decide how you have modified the surface by means of printing or by means of treatment.

Answer: In AM, 3D-printed objects are designed with porous structures providing the geometry of the printed object. The organic acid or electropolishing treatment are made to remove the particles of titanium powder adhered to the specimen surface or internal pores after sintering procedure. The acid treatment does not modify the porosity decided by CAD design but by removing the particles it generates a micro-roughness of the surface. The electropolishing procedure is used to obtain smoother surface.

  • In Fig.1 the scale bar is missing.

Answer: I apologize, the scale bars have been added.

  • In Fig.2 and Fig.4c the scale bars are not readable, please fix that.

Answer: the scale bars have been fixed, as requested.

  • lines 43-44 "Among the methods of AM, selective laser melting (SLM) and sintering (SLS) have been the predominant methods for the AM of titanium alloys." - please provide a reference for this statement. First, what about Electron Beam melting? Second, SLS and SLM are the same technology. The main difference between them is the fusing process - or sintering or melting. Sintering (SLS) is used for nylons/polymers; melting (SLM) - for metal alloys. The term "SLS" is a historical name of the same powder bed technology. Now, for titanium alloys, only SLM is used, not SLS. Please, fix this part in Introduction (lines 43-48). I also recommend to observe some relevant papers on this topic, e.g.: Answer: This part within the introduction section has been fixed and better explained, as requested by the referee. Moreover, references 2 and 3 have been replace by more appropriate ones (https://doi.org/10.1007/s13534-018-0080-5; https://doi.org/10.1016/j.matdes.2017.07.059) and the reference list corrected accordingly.
  • line 71 "MSCs" - What does it mean? Please give an explanation of all abbreviations.

Answer: Corrected.

  • lines 78-79 "Furthermore, we have recently reported..." - you can't write like that since there were other authors in [16].

Answer: we apologize, the error has been corrected as requested.

  • lines 91-93 "The additive manufacturing (AM) SLM technology is based on the process called metal powder bed fusion in which material powders are fused together [17]." - IT is not relevant for Methods and Materials Section. Please, or remove, or put that in Introduction.

Answer: the sentence and the cited reference have been removed.

  • line 99 "After that, the laser melted samples where heat-treated at 800°C for 1 hour" - in what environment? Vacuum/Argon?

Answer: the treatment has been performed in Vacuum heat; the information has been added in the text.

  • Section 2.1. synthesis of the samples types A, B, and C. I recommend transforming this section into a representative table indicating process parameters, and applied post-processing procedures.

Type A - Material - Size/shape - synthesis - heat treatment - post-processing

Type B ....

Type C ...

Answer: we thank the referee for his suggestion. The information regarding sample parameters and applied post-processing procedures have been also placed within a table (named table 1) at the end of paragraph 2.1.

  • Just a short explanation is required why you have used different printing machines for different types of samples? Answer: The two companies that produced the samples have different printing machines and manufacturing procedures.
  • And why do you show different process parameters e.g. layer thickness is shown only for the Sisma machine?

Answer: We apologize, we added the layer thickness for the Renishaw machine which is 15 μm.

  • Please, specify, the powder data - supplier, morphology, particle size dispersion, production technique, chemical composition, etc.

Answer: Powder used for RenAM 500Q and MYSINT 100 printers is the same (produced by Carpenter Additive - Dennis Rd, Widnes WA8 0GU, United Kingdom). PowderRange Ti6-4 Grade 23 (Titanuim-6%, Aluminum-4%, Vanadium, Ti-6Al-4V) is a high-performance alloy with excellent mechanical properties,with a low specific weight and good corrosion resistance. It has lower specified oxygen, carbon and nitrogen limits, and is considered the higher purity version. Grade 23 displays increased ductility and fracture toughness over Grade 5 as a result of the reduction in interstitials, as well as demonstrating excellent biocompatibility. The powder is produced by plasma atomization, resulting in superior sphericity, low internal porosity and low residual elements.

All the information has been added at the beginning of the first paragraph in Materials and Methods section. Below we report also a table with further information regarding the powder composition, if the referee wants we can include the table within the paper.

Chemical composition Element

Minimum wt%

Maximum wt%

Al

              Aluminum

5.50

   6.50

C

Carbon

                 0.08

H

Hydrogen

                 0.012

Fe

Iron

                 0.25

N

Nitrogen

                 0.05

O

Oxygen

                 0.13

Ti

Titanium

              Balance

V

              Vanadium

3.50

4.50     

- line 412 - just "melted", no need to say each time "melted and fused"; lines 411-418, please try to avoid such characteristics as "small powders" or "very rough surface". These characteristics are measurable. The standard size range for SLM powders is 25-50 microns. The standard roughness is related to the powder you use, so for SLM it also will be of the order of 50 microns. Anyway, this paragraph is more suitable for Introduction not for a discussion.

Answer: the sentence has been removed as recommended.

  • line 116 and 424 "surface roughness" for which purpose you put a link on Wikipedia here???

Answer: we apologize, it was a mistake, it has been corrected.

  • lines 419-425 - it is repeated information, please remove.

Answer: the information has been removed as requested.

  • line 426 "was" instead of "were"

Answer: corrected.

  • Conclusion: "On the other hand, pulp cells grown on type A titanium show a better profile in terms of markers related to osteogenesis, a cell behaviour plausibly ascribable to its open cell inner structure, which resembles the native bone trabecular structure." Please, explain why it is preferable from your point of view to treat the surface of titanium implants, instead of just printing from the beginning the trabecular surface [https://doi.org/10.1007/s13534-018-00092-7]?

Answer: As explained, the organic acid or electropolishing treatments are made to remove the particles of titanium powder adhered to the specimen surface or entrapped into pores after sintering procedure. The acid treatment does not modify the porosity decided by CAD design but by removing the particles it generates a micro-roughness of the surface. The electropolishing procedure is used to obtain smoother surface. The study wants to highlight that porous structure of titanium drawn in CAD and produced by laser melting (SLM) demonstrates greater bioactivity than dense surfaces. This sentence was added within the conclusion section.

  • I recommend to strengthen the Conclusion and maybe present the main findings in the form of a bullet list.

Answer: the conclusion section has been reinforced by adding and strengthing some information.

  • The paper has serious gaps, and I recommend to work on Abstract, Introduction summarizing part, and on Conclusion, to formulate the novelty and value of the presented research.

Answer: we thank the referee for all the suggestions, the manuscript has been overall modified focusing in particular on the sections recommended by the referee.

  • I also don't understand the classification of the investigated samples: A - titanium with porous structure; B and C without. Moreover, you apply different treatments to A and to B/C. That means you simultaneously compare the effect of porosity and treatment, and it is not good. More reasonable to compare samples type A as-built, heat-treated, and post-processed with different cell structures. Thus, you can compare the initial biocompatibility of the additively manufactured porous titanium and the biocompatibility of the samples with a treated surface. Otherwise, your experiment design is not clear.

Answer: The main purpose of the article is to compare porous titanium structures with dense structures and their biocompatibility and osteoconductive features in in vitro models represented by Dental Pulp Stem Cells (DPSCs). We described the manufacturing process and the surface cleaning process to highlight all the differences between samples A (porous titanium) and samples B and C (dense titanium).

Reviewer 2 Report

  1. On the basis of your mention in line 246, How did you know the interconnected pores in the sample Ti A? If you want to write as like that, you should check the cross-sectional view in terms of morphology with SEM.
  2. In line 270, you just only mentioned the metabolic activity slightly decreased after 28 days. Why that kinds of phenomena was come out? I think you should explain more the reason or mechanism.. 

Author Response

  • On the basis of your mention in line 246, How did you know the interconnected pores in the sample Ti A? If you want to write as like that, you should check the cross-sectional view in terms of morphology with SEM.

Answer: We checked the cross-sectional view of the specimens surface but for reasons of brevity we have not included the photo (file number 20007_11) within the manuscript, that we attach separately. If the referee wants we can include the image in the paper.

  • In line 270, you just only mentioned the metabolic activity slightly decreased after 28 days. Why that kind of phenomena was come out? I think you should explain more the reason or mechanism.

Answer: Thank you for the suggestion, we added the explanation of this aspect within the following paragraph which has been added in the discussion section. A new reference note has been added also to better support our results (reference n. 23)

“The osteogenic commitment of MSCs is characterized by a high proliferation rate and a change in cell energy metabolism. It has been reported that many types of stem cells rely on glycolysis for energy when undifferentiated, and then later activate the heavily mitochondrial process of oxidative phosphorylation (OxPhos) during differen-tiation leading to a highest metabolic state [Shum L.C. et al., Stem Cells Dev. 2016, 25(2):114-22]. This is in line with our experimental data related to the Alamar blue assay up to 21 days, where DPSC proliferation is clearly increased. However, after 28 days, the proliferation of differentiated cells slightly de-creases. As broadly known, osteoblasts and osteoblast-like cells are mature and highly differentiated post-mitotic cells whose function is to produce the mineralized matrix and thus characterized by a low proliferation rate and an augmented protein synthesis. This trend has been already reported by our group related to established cell/biomaterial experimental models in the presence of DPSCs [Rapino M. et al., Nanomaterials (Basel). 2019, 9(7):928].

Round 2

Reviewer 1 Report

The authors have significantly improved the representativeness of the article and fixed the major issues.
I still suppose that the Conclusion can be improved, as I said previously, to highlight the main findings in a form of a bullet list. But, anyway, even in a current form the manuscript is already publishable. 

Author Response

Thank you for you valuable revision.

Reviewer 2 Report

I think your supplemental data shows the three-dimensional connect structure.

Author Response

Thank you for the suggestion to add this supplemetal image, we will provide the related caption. 

English style has been checked again and improved.